# An Application of a Hybrid Intelligent System for Diagnosing Primary Headaches

**DOI:** 10.3390/ijerph18041890

**Published:** 2021-02-16

**Authors:** Svetlana Simić, José R. Villar, José Luis Calvo-Rolle, Slobodan R. Sekulić, Svetislav D. Simić, Dragan Simić

**Affiliations:** 1Faculty of Medicine, University of Novi Sad, 21000 Novi Sad, Serbia; svetlana.simic@mf.uns.ac.rs (S.S.); slobodan.sekulic@mf.uns.ac.rs (S.R.S.); 2Faculty of Geology, Campus de Llamaquique, University of Oviedo, 33005 Oviedo, Spain; villarjose@uniovi.es; 3Department of Industrial Engineering, University of A Coruña, 15405 Ferrol-A Coruña, Spain; jlcalvo@udc.es; 4Faculty of Technical Sciences, University of Novi Sad, 21000 Novi Sad, Serbia; simicsvetislav@uns.ac.rs

**Keywords:** intelligent system, headaches, analytical hierarchy process, fuzzy c-means clustering

## Abstract

(1) Background: Modern medicine generates a great deal of information that stored in medical databases. Simultaneously, extracting useful knowledge and making scientific decisions for diagnosis and treatment of diseases becomes increasingly necessary. Headache disorders are the most prevalent of all the neurological conditions. Headaches have not only medical but also great socioeconomic significance. The aim of this research is to develop an intelligent system for diagnosing primary headache disorders. (2) Methods: This research applied various mathematical, statistical and artificial intelligence techniques, among which the most important are: Calinski-Harabasz index, Analytical Hierarchy Process, and Weighted Fuzzy C-means Clustering Algorithm. These methods, techniques and methodologies are used to create a hybrid intelligent system for diagnosing primary headache disorders. The proposed intelligent diagnostic system is tested with original real-world data set with different metrics. (3) Results: First at all, nine of 20 attributes – features from International Headache Society (IHS) criteria are selected, and then only five most important attributes from IHS criteria are selected. The calculation result based on the Calinski–Harabasz index value (178) for the optimal number of clusters is three, and they present three classes of headaches: (i) migraine, (ii) tension-type headaches (TTHs), and (iii) other primary headaches (OPHs). The proposed hybrid intelligent system shows the following quality metrics: Accuracy 75%; Precision 67% for migraine, 74% for TTHs, 86% for OPHs, and Average Precision 77%; Recall 86% for migraine, 73% for TTHs, 67% for OPHs, Average Recall 75%; F_1_ score 75% for migraine, 74% for TTHs, 75% for OPHs, and Average F_1_ score 75%. (4) Conclusions: The hybrid intelligent system presents qualitative and respectable experimental results. The implementation of existing diagnostics systems and the development of new diagnostics systems in medicine is necessary in order to help physicians make quality diagnosis and decide the best treatments for the patients.

## 1. Introduction

A great deal of information in the field of modern medicine is stored in medical databases. These databases have also become necessary for the extraction of useful knowledge and scientific decision making for the diagnosis and treatment of diseases. The medical field is concerned with patient care activity primarily and research only secondarily. Thus, the justification for the collection of medical data is established on the benefits for individual patients. 

Headache disorders can be considered as the predominant neurological condition and one of the most frequent medical complaints in general practice. More than 90% of the general population report suffering from a headache during any given year, which can be regarded as a lifetime history of head pain [1], while over 10% have had at least one migraine headache [2]. Headache is not a disease which typically shortens one’s life, but it can be a serious social as well as a health problem. A lot of people with headache are unaware of the type of headache they are suffering from. 

Headache is one of the most frequent complaints in medicine in general and a frequent disorder in the working population that significantly affects absenteeism and loss of productivity [3,4,5]. The impact of headache disorders is a problem of enormous proportions, both for individuals and the society. Different diseases, including headaches, which may develop into disabilities, were subjects to investigations mostly in the developed countries, regarding the effects of illness on absenteeism and the reduced productivity [6,7]. The cost–benefit and risk–benefit ratios are two of the most relevant items in ongoing health-organization procedures [8]. In the field of neurological diseases, migraine was estimated to cost a total of €27 billion per year for the loss through reduced work productivity in the European Community. The second most costly neurological disease was stroke, totaling €22 billion, followed by epilepsy, Parkinson’s disease and multiple sclerosis with the costs of €16, €11 and €9 billion, respectively [9]. Migraine is highly disabling, associated with substantial economic burden. Approximately 13 billion dollars per year are lost through reduced work productivity in the United States [10]. 

Therefore, physicians frequently find themselves in situations where they have to consider the economic aspects of a treatment, even though they have not been trained for it. Healthcare requires making decisions about diagnosis and treatment that are compatible with current medical opinions and practices along with taking care of its availability and patient compliance to treatment [11,12]. On the other side, people do not always visit physicians for their complaints, and when they do, physicians do not always make a correct diagnosis. The education of both health professionals and patients is also significant. Better education means better treatment results and less economic loss [13]. The epidemiological research [14] has heavily depended on the diagnostic criteria introduced by the International Headache Society (IHS), and some automatic methods, expert systems, and knowledge-based systems, including some tools that help physicians make diagnoses, have been developed and based on these criteria.

Therefore, according to the previous facts, this research is focused on computerized-based application of a hybrid intelligent system for diagnosing primary headaches. In this research, the clustering method in general and the fuzzy c-means clustering approach in particular are used. The main reason is the fact that in the real-world setting, the physicians do not know the type of primary headache in advance. Therefore, it is necessary to use some artificial intelligent technique which satisfies this objective. Likewise, some other mathematics, statistics, and decision-making techniques are applied to create an intelligent system for diagnosing primary headache disorders. The proposed intelligent system is tested on the real-world data set collected from the Clinical Centre of Vojvodina, Serbia. This paper presents a research that extends from and is applied on the basis of our previous research [15] and on the research in computer-assisted diagnosis methods [16,17,18], as well as on the industrial applications developed for clustering methods [19,20].

## 2. Materials and Methods

### 2.1. Primary Headache Classification

The uniform terminology, as well as the consistent operational diagnostic criteria applied to a wide range of the headache disorders around the world [21] are established on The International Classification of Headache Disorders–The Third Edition (ICHD-3). The ICHD-3 offers a hierarchy of diagnoses specifying a number of varying degrees. Headache disorders are usually identified using three-digit codes, and sometimes using five-digit codes, which is displayed in detail in Table 1 with the short identification for only two important digit codes.

All headache disorders are classified into two major groups: (A) *Primary headaches*, from ICHD-3 code 1. to 4., and (B) *Secondary headaches*, ICHD-3 code from 5. to 12. The first digit specifies the major diagnostic categories (i.e. migraine). The second digit indicates a disorder within the category (i.e. migraine without aura). Each category is then subdivided into groups, types, subtypes and sub-forms. Subsequent digits permit more specific diagnosis for some headache types [21]. 

### 2.2. Data Extraction

Our previous study was completed on a sample of 1022 employed subjects, who filled in the questionnaire in their workplace. The statistical analyses of that research are presented in [15]. The experimental results show that, out of the 1022 subjects who completed the survey, 579 (56.6%) reported headaches; 169 (16.5%) had migraines, 224 (21.9%) had tension-type headaches (TTHs), and 186 (18.1%) had other headache types. The category of other headache types included patients with a headache that, according to their answers, could not be classified as a migraine or TTHs. 

The previous research [15] is continued with the idea to create a computerized-based intelligent system for diagnosing primary headaches to assist and help physicians make correct diagnosis for the patients who suffer from primary headache disorders and their treatments. This research was approved by the Ethics Committee of the Clinical Center of Vojvodina on 4 June 2020 (no. 00-214). This research includes only subjects with headache disorders, who were further analyzed. This experiment in total used the data set from 579 instances (subject–patients) and 20 features–attributes (answers to questions in the survey). The data set is divided in to three classes (groups of headaches), which included the following types of primary headache: Migraine, Tension Type Headache (TTH), and Other primary headaches (Other); where the missing data does not exist.

The previous study examined a number of studies based on IHS recommendations [22]. Different approaches were presented to attribute to the selection based on automatic methods, knowledge–based systems, expert systems, as well as physicians’ expert knowledge, as displayed in Table 2. The selection of features could be separated into *Stochastic* and *no-Stochastic* Feature Selection methodology, which presents a refinement of the initially used stochastic feature selection task with a no-stochastic method in order to further reduce the subset of features that were to be retained [23]. The previous study demonstrates that the most important features are the following: A4, A5, A6, A7, A8), A10, A12), A13 and A15; these are presented in Table 2 in black bold font. The emphasized nine features are used in the remain of the research [22].

### 2.3. Early Literature Research 

During the last several decades, a sufficient number of approaches have been proposed to explain the clustering problem in medical data with the aim of helping physicians make decisions related to patient’s illness and their future treatments. One of the first studies, which contained 726 headache patient analyses, dates back to 1982 and it is presented in [24]. In that study, the cluster analysis was used as a method to find groups (clusters) of patients with similar symptoms. When only two clusters were required, the best two clusters were TTHs and migraine-like. However, in that research, eight clusters could also be distinguished, and the migraine group then became very small. 

Based on previously mentioned, it can be concluded that the first step in the clustering problem, defining the number of clusters, is a very complicated question with the uncertain answer. After forming the clusters, the frequency of every symptom in every cluster was tabulated. Twelve physicians (eight internists, a cardiologist, a pediatrician, a neurologist, and a pathologist) were asked to use these symptom frequencies in order to label each cluster and then to prescribe what would a typical therapy be for every cluster. The “IF–THEN” rule algorithm was developed and subsequently tested, proving that 92.3% headache sufferers were placed in the correct cluster.

### 2.4. Optimal Number of Clusters

The Calinski-Harabasz index, as the concept of dense and well-separated clusters, is used to estimate the optimal number of clusters. Two measures were used, the Variance Ratio Criterion and the Total within Sum of Squares, in order to select the suitable *c*, i.e., a number of clusters. For creating the Calinski-Harabasz index, it was required to define the inter cluster dispersion [25] first. When *N*, i.e., the total number of observations (data points), and *c* number of clusters with their relative centroids and the global centroid, are known, the inter-cluster dispersion *B*(*c*) (between cluster variations) can be defined as follows:(1)$$B(c)\;=\;\sum_{t=1}^{C}n_{t}\;{\left(\mu \;-\;\mu_{t}\right)}^{T}\;(\mu \;-\;\mu_{t}),$$

In Equation (1), *n_t_* is the number of elements that belongs to the cluster *c*, *µ* is the global centroid, *µ_t_* is the centroid of the cluster *t*. 

The intra-cluster dispersion *W*(*c*), within cluster variation, is defined as:(2)$$W(c)=\;\sum_{t=1}^{C}\sum_{x\in C_{N}}^{N}{\left(x-\mu_{t}\right)}^{T}(x-\mu_{t}),$$

The *Calinski-Harabasz index* is defined as the ratio between *B*(*c*) and *W*(*c*):(3)$$CH(c)\;=\;\frac{N-c}{c-1}\;\times \;\frac{B(c)}{W(c)},$$

The *Calinski–Harabasz index* is based on the comparison of the weighted ratio of the between cluster sum of squares (the cluster separation measure) and the within cluster sum of squares (the measure presenting how points are tightly packed within a cluster). The number of clusters maximizing this index needs to be recovered in order to determine low intra-cluster dispersion and high inter-cluster dispersion. Ideally, the clusters should be well separated, and the between cluster sum of squares value should be large. On the other hand, points within a cluster should be as close to one another as possible, which would result in smaller values of the within cluster sum of squares measure [25]. The decision that a point will be assigned to a cluster depends only on its features and sometimes on the position of a set of other points. Likewise, there are diverse algorithms based on alternative strategies to solve this problem, which can yield diverse results [26].

### 2.5. Analytical Hierarchy Process

Analytic Hierarchy Process (AHP) is one of multi criteria decision making methods that was originally developed and first time presented in [27]. It is a method to derive ratio scales from paired comparisons. The AHP starts by decomposing a complex, multi-criteria problem into a hierarchy where each level consists of a few manageable elements (group criteria *C_i_*) which are then decomposed into another set of elements (criteria *C_ij_*). Later, these criteria (*C_ij_*) are mutually compared in order to get the priority of each criterion in hierarchy. Finally, all alternatives are compared in relation to the set of criteria (*C_ij_*) and in this way the comparison of alternatives (*A_i_*) is obtained. The detail description of AHP flowchart is shown in Figure 1 [28]. 

In order to get the final result and in order to get the gradation of influence for each criterion, first it is necessary to compare criteria mutually. The comparison of any two criteria *C_i_* and *C_j_* with respect to the goal is made using the questions of the type: of the two criteria *C_i_* and *C_j_* which is more important and how much. In reality, however, an individual cannot give estimates such that they would conform to perfect consistency. It means that the final result might be accepted when consistency ratio (CR) is <0.1, less than 10%. Otherwise, the problem has to be revised [29]. 

### 2.6. Weighted Fuzzy C-means Clustering Algorithm

*Fuzzy c-means* (FCM) *clustering algorithm* is one of the most popular fuzzy clustering techniques, which was proposed in [30] and partially modified in [31]. It is an approach where the data points have their membership values with the cluster centers (centroids), which will be updated iteratively. In this research, the classical FCM algorithm is extended with the weight matrix of the features. The basic steps of the proposed Weight FCM clustering algorithm, employing a Euclidean distance norm, are summarized by the pseudo code shown in Algorithm 1.
**Algorithm 1:** Weighted fuzzy c-means clustering algorithmBeginStep 1:---Initialization.X, c, ε > 0, WStep 2:---Randomly select V cluster centers.2 ≤ c ≤ NStep 3:---Choose an appropriate level of cluster fuzzinessf.f [1, ∞], f > 1Step 4:---Choose an appropriate membership matrix U with size N × c × M*U_ijm_* ∈ [0, 1] and ∑j=1cUijm = 1 for a fixed value of mStep 5:---Calculate the cluster centers.Repeat for *j*th cluster and its *m*th dimensionCCjm = ∑i=1NUijmf xim wm∑i+1NUijmfStep 6:---Calculate the Euclidean distanceDijm = ‖(xim wm− CCjm )‖Step 7:---Update fuzzy membership matrix U according to *D_ijm_*Uijm = 1∑c=1C(DijmDicm)2f − 1Step 8:Until U ≤ εEnd.

Let us suppose *X* = {x_1_, x_2_, x_3_,..., x*_N_*} be the set of data points, where *N* is the number of data points, in M-dimensional (M = 1, 2,..., m) data space. Let *V* = {*v*_1_, *v*_2_, *v*_3_,..., *v_c_*} be the set of cluster centers, and *c* represents the number of cluster centers. Let *W* be M-dimensional weighted matrix of the features. Let *ε* be the termination tolerance > 0 where termination criterion is between [0, 1] (Step 1). Assume the number of clusters *c* to be made, where 2 ≤ *c* ≤ *N* (Step 2). Choose an appropriate membership matrix *U* with size *N* × *c* × *M*, and conditions which are presented in Step 4. Determine the cluster centers *CC_jm_*, including weighted matrix of the features, for *j*th cluster and its *m*th dimension by using the expression in Step 5. Calculate the Euclidean distance *D_ijm_* between *i*th data point and *j*th cluster center with respect to *m*th dimension (Step 6). Update fuzzy membership matrix *U* according to *D_ijm_*, if *D_ijm_* > 0 (Step 7). Finally (Step 8), *Repeat* from Step 5 to Step 7 *Until* the changes in *U* ≤ ε, ε is a pre-specified termination criterion.

### 2.7. Implemented System for Diagnosing Primary Headaches 

Figure 2 depicts the proposed hybrid intelligent system used for the diagnosis of primary headache disorders implemented in the research. The system comprises of two phases. The first phase is realized in four steps: (1) define and select nine important attributes of input data set based on our summarized research presented in Table 2, called new data set; (2) estimate the optimal and correct number of clusters calculated by Calinski–Harabasz index value; (3) according to previous estimation, select five most important attributes from new data set based on means and standard deviation, called Selected data; (4) apply Analytical Hierarchy *Process* to determine weighted coefficients for attributes in *Selected data*. It could be concluded that all activities in the first phase are focused on preparing the original data set with selection attributes and their weighted coefficients for implementation, which is called Define attributes and weights. 

In the second phase, FCM clustering method is applied, and called Fuzzy Method and Decision. After fuzzy c-means clustering method, *Solution* presents three clusters with three cluster centers in five-dimensional data space. Three clusters display Diagnosed Patients with Primary Headache with adjoining types which suffer of: migraines, TTHs, and other primary headaches. 

### 2.8. Metrics

In this research, the focus is on the multi-class problem. The confusion matrix for the multi-class problem is presented in Table 3.

TP (true positive) stands for the number of correctly classified positive instances. The existing elements in the form E_XY_ present *true negative*, which stands for correctly classified negative examples, or false positive, as the numbers of incorrectly classified objects from classes. On the basis of the confusion matrix, the following metric can be calculated, showing the quality of the correct predictions. Accuracy is one of the most popular thresholds metrics. This is a measurement of the correct predictions between all predicted and actual values. *Accuracy* for the multi-class problem is presented by Equation (4):(4)$$Accuracy\;=\;\frac{\sum_{i}TP_{i}}{\sum_{i}TP_{i}\;+\;\sum_{i}\sum_{j}E_{ij}},$$

The other popular thresholds metrics, which will be defined for the multi-class problem, are: *precision* (Equation (5)), *recall* (another term for *Sensitivity*) (Equation (6)) and *F*_1_
*score* (Equation (7)):(5)$$precision{\;}_{i}\;=\;\frac{\;TP_{i}}{\sum_{j}E_{ji}},$$
(6)$$recall{\;}_{i}\;=\;\frac{\;TP_{i}}{\sum_{i}E_{ij}},$$
(7)$$F_{1}\;score\;=\;2\;\times \;\frac{precision\;\times \;recall}{precision\;+\;recall},$$

These appropriate metrics will be used for measuring quality for the proposed hybrid intelligent system for diagnosing primary headaches.

## 3. Results

The *Input Data Set* consists of 579 instances (patients); 169 of them have migraines, 224 have TTHs, and 186 have another primary headache type. Following the flowchart for the hybrid intelligent system for diagnosing primary headache presented in Figure 2, the first step in the first phase is defining nine important attributes. The important features are: A4, A5, A6, A7, A8, A10, A12, A13 and A15 presented in Table 2 and discussed in details in [22]. 

The next step is calculating *Calinski-Harabasz index* value presented in Figure 3. The *Calinski-Harabasz index* value is as follows: for two clusters it is 189; for three clusters it is 178; for four clusters it is 165; for five clusters it is 148; and for six clusters it is 123. As mentioned before, whole data set will be clustered in three classes; therefore, it is decided for *Calinski-Harabasz index* value to be 178 and the optimal number of clusters to be three (Figure 3).

In the next step in phase one, five most important attributes based on the diversity means and standard deviation values in three clusters are selected, and presented in Table 4. It could be concluded, after reducing the attributes, that the most important attributes are: A4, A6, A7, A8, and A15. These attributes describe the following characteristics of headache types: A4-length of headache attacks; A6-intensity of the pain; A7-quality of the pain; A8-headache is worsen after physical activities; andA15-tension or heightened tenderness of head during a headache attack.

In the last step of the first phase, AHP method is included. The pairwise comparison matrix, which compares the priority of the attributes, is presented in Table 5.

The consistency ratio (CR) is CR = 0.0174, which is less than 0.1, and pairwise comparison matrix is satisfied. Then, the priority weights for the attributes are: Attribute (4) is 0.2260; Attribute (5) is 0.1530; Attribute (6) is 0.2260; Attribute (8) is 0.1975; and Attribute (15) is 0.1975:

*Priority Weights* = [0.2260; 0.1530; 0.2260; 0.1975; 0.1975] 

In the second phase, weighted phase fuzzy c-means clustering algorithm is applied. The confusion matrix for actual classes and predicted classes (Migraine, TTHs, Other) headaches is presented in Table 6.
*Accuracy* = (146 + 163 + 125)/579 = 0.7496,(8)

Now, the Accuracy for the whole Hybrid Intelligent System for Diagnosing Primary Headaches can be calculated, and it is presented with Equation 4. Accuracy for the Hybrid Intelligent System for Diagnosing Primary Headaches is 0.7496 Equation (8), which is 75.0%, presenting a qualitative and respectable result. 

The quality measurement for the proposed hybrid intelligent system for diagnosing primary headache is presented in Table 7. The appropriate metrics measuring qualities are: precision, recall and F_1_ score. In the multi-class problem, *precision*, *recall*, and *F_1_ score* are presented for every attribute–feature separately, and *average* values for every metrics are calculated. It is a different situation compared to metrics in two-class problem, when only one value for every metrics, precision, recall, and F_1_ score, is defined.

The implementation of the hybrid intelligent system for diagnosing primary headaches presents qualitative and respectable experimental results. The average of all metrics values is about 75%, which presents a qualitative result. 

## 4. Discussion

The discussion in this section will be mostly focused on the usage of different clustering algorithms and decision support systems and their application in the diagnosis of primary headaches. The total number of 150 headache-prone subjects responded as follows: there were 49 migraines and 101 TTHs examined [32]. A cluster analysis was introduced, and the adjectives were thus grouped into seven clusters, which included five sensory and two affective sub-groups. Headache was most commonly described in relation to clusters for reflecting discomfort and aching pain sensations. There was not any marked difference in pain quality referred by migraine and TTH sufferers; however, the intensity of pain differentiated between these groups. In order to examine the relationship between descriptors, a complete linkage cluster analysis as one of several methods of *agglomerative hierarchical clustering*, was performed. Experimental results demonstrated the correspondence between migraine and TTH in two new cluster groups which were derived after using the cluster analysis: migraine, 93 of 101 (92.1%); TTH, 49 of 49 (100%); and misclassified, eight (5.3%) patients; thus, the diagnosis average accuracy is 94.7%.

A new proposed migraine analysis method was based on using electroencephalography signals under flash stimulation in time domain. These types of signals are normally pre-processed before the analysis procedure, where the pre-processing techniques affect the analysis results. Histogram differences in the case of flash stimulation are calculated and the results are used as features for the healthy subjects and migraine patients. These features are also applied to a *k-means* clustering algorithm in order to be able to observe clustering results of the proposed technique. *Silhouette clustering* results demonstrate that a good clustering performance is evaluated as 86.6% of the correct clustering rate in migraine patients [33].

The *ant colony optimization* classification algorithm is used for the diagnosis of primary headaches by using a website questionnaire expert system; it has shown the overall diagnosis accuracy as 96.9% [34]. In order to evaluate diagnosis accuracy of *artificial immune system*, algorithms for the classification of migraine, TTH and cluster headache have the maximum accuracy of 71% [35].

A hybrid fuzzy clustering approach for diagnosing primary headache disorder which combines fuzzy partitioning method and maximum likelihood estimation clustering algorithm is presented in [36]. *Calinski–Harabasz index* value is used to estimate the optimal and correct number of clusters. The proposed hybrid system is tested on the data set which has 579 patients who suffer from headaches, and it is facilitated by the application of the IHS criteria for diagnosing primary headache disorder. The headaches are clarified in four classes: migraine without aura (MWoA), migraine with aura (MWA), TTHs, and Other primary headaches. There are pairwise comparisons used between classes, where the minimum accuracy value is 78.1% and the maximum accuracy value is 86.4%. The *Average Accuracy* of that applied fuzzy clustering system is 82.9%. 

Decision Support System (DSS) application determines the type of Headache using *Fuzzy Multi Attribute Decision Making* (FMADM) *Method* with *AHP*, as one of the methods that assists decision makers in making decisions; it is considered in [37]. A triangular membership function in fuzzy process is selected and used. The *fuzzy compatibility index* (FCI) is calculated for three types of headaches: FCI = 62.5% for migraines, FCI = 77.5% for TTHs, FCI = 57.5% for cluster headache.

The diagnostics system that was developed in Italy in 1998 is known as “IHS *Diagnostic Criteria for Primary Headache*”. The system was developed strictly based on the IHS operational diagnostic criteria. The system was tested on the computerized structured record in the cases of 500 patients in nine headache centers in Italy and the experimental results are depicted in [38]. The rate of concordance was calculated between the diagnosis provided by the computerized structured record and the one reported by physicians. The concordance between two diagnoses was recognized in 345 of 500 cases examined (69%). Further software implementation from 2004 was focused on 200 patients who reported primary chronic headaches. A certain diagnosis of a chronic form and a probable form of headaches was obtained in 50.8% cases [39].

In [40], a computerized program designed to diagnose primary headache was developed in Iran and tested in Tehran (Iran). It was based on International Classification of Headache Disorders, 2nd edition (ICHD-II) criteria which was to be used by physicians. A total of 80 patients were tested with the developed system. The software tool was able to come up with correct results in 78 out of 80 cases. Migraine headache accounted for 71 cases, five patients had TTHs, and two had cluster headaches; all were correctly diagnosed by the software. Two cases were not concordant with the neurologist’s diagnosis which represents an *Average Accuracy* of 97.5%.

The experimental results from this research are comparable with researches discussed in this Section, as well as with other researches. The Average Accuracy in this research is 75.0%, which is in the middle compared to other studies where Minimum Accuracy is 57.5% and Maximum Accuracy is 94.6%. On the other side, it is important to mention that in many papers methodologies are presented, but in fewer papers the detailed steps are provided, and only in a very small number of papers, are detailed steps of the methodology and intermediate experimental results presented, as is the case in our study. In other research papers, final experimental results are mainly presented without intermediate experimental results; therefore, it may be a bit difficult to follow the complete applied methodology. 

Conversely, a number of applied statistical methods in the diagnosis of primary headache disorders is used for analyzing different features and variables. For example, descriptive statistics with categorical data presented using frequencies and percentages can be used for presenting continuous descriptive data that uses means and standard deviations, which can in turn be compared using the Chi-square test or Fisher’s exact test in order to compare two groups of categorical variables. Then, the Student ‘t’ test may be used, as well as Mann–Whitney ‘U’ test for continuous variables, or logistic regression analysis to determine binary dependent variables [41,42,43,44].

Regarding the socioeconomic impact on society, the studies have evaluated both the indirect costs of migraine as well as the direct costs [45,46,47]. The indirect cost of pain may be estimated with the number of missed work days. Indirect costs include the aggregate effects of migraine on productivity at work (paid employment), on household work, and in other roles [48]. Working people with migraine dropped out of work 14 days per year, which adds up to 6.8 million working days per year in Austria [49]. This remains a substantial economic factor. Migraine is estimated to account for about 1,154,336 lost working days each year in Flanders and Brussels [50]. The largest component of indirect costs is the productivity loss caused by absenteeism and reduced productivity; it is estimated that productivity losses due to migraine cost American employers 13 billion dollars per year [45]. A European study estimated that 5.7 working days were lost per year for every working or student migraineur (although the most disabled 10% of migraineurs accounted for 85% of the total), which projects to a loss of 25 million days from work or school each year in England [51]. A realistic, rational, and economical approach with adherence to the current guidelines for diagnosis and treatment is almost an impossible task for a practicing physician. In less developed countries, such as Serbia, this problem is even greater [13]. Therefore, it is necessary to research new automatic methods, intelligent and expert systems, or knowledge–based systems to help and improve physicians work to make better diagnoses and treatment for the patients. 

Weaknesses in diagnosis occur in everyday clinical work, since atypical disease forms exist, and such cases are certainly present in our research as well. Likewise, the characteristics of the disease - which are presented as attributes in this study, do not mean that each observed case meets all the characteristics in 100%, as established in The International Classification of Headache Disorders–Third Edition (ICHD-3) [21]. The difference in clinical presentations in some cases leads to deviations from the recommended criteria in ICHD-3, which is a weakness of the recommended hybrid intelligent system for diagnosing primary headaches. On the other hand, the patient may suffer from several types of headaches that occur at different times. Therefore, the obtained answers, i.e. attributes, can be a combination of two types of headaches, leading to discrepancies in the proposed diagnosis provided by the intelligent system and the diagnosis by a neurologist, representing their expert assessment. The occurrence of two types of headaches with the same patient can be interpreted as the mean value of the observed attributes, having similar values for different types of headaches, as displayed in Table 4 for attributes A5, A10, A12, and A13 for TTHs and Other headache types. 

The advantage of this system is the identification of the attributes that are dominant for different types of headaches. In this hybrid intelligent system for diagnosing primary headaches, their significance, in relation to other attributes, is increased by weighting coefficients in weighted fuzzy c-means clustering algorithm. The goal of our research is to help physicians have a reliable tool that would help them make the better diagnosis of primary headaches.

It can be concluded that the main contributions of our paper are: The application of the *Hybrid Intelligent System for Diagnosing Primary Headaches* and computational complexity calculation.Experimental evaluation of the application of the *Hybrid Intelligent System for Diagnosing Primary Headaches* and comparison with the selected state-of-the-art methods.

## 5. Conclusions

Headache disorders are the most prevalent of all neurological conditions. The goal of this research is focused on the design and implementation of a computer-based hybrid intelligent system for diagnosing primary headaches. This research applied various mathematical, statistical and artificial intelligence techniques, including decision-making methodology and clustering methods. The proposed hybrid intelligent system for diagnosing primary headaches is tested on real-world data set collected from the Clinical Centre of Vojvodina (Serbia). The *Accuracy* of the proposed intelligent system is 75.0%, which is a respectable value and a value comparable with similar research. Further research could be focused on creating new and more efficient tools and systems to help and improve physicians’ work and make diagnoses better. 

## Figures and Tables

**Figure 1 ijerph-18-01890-f001:**
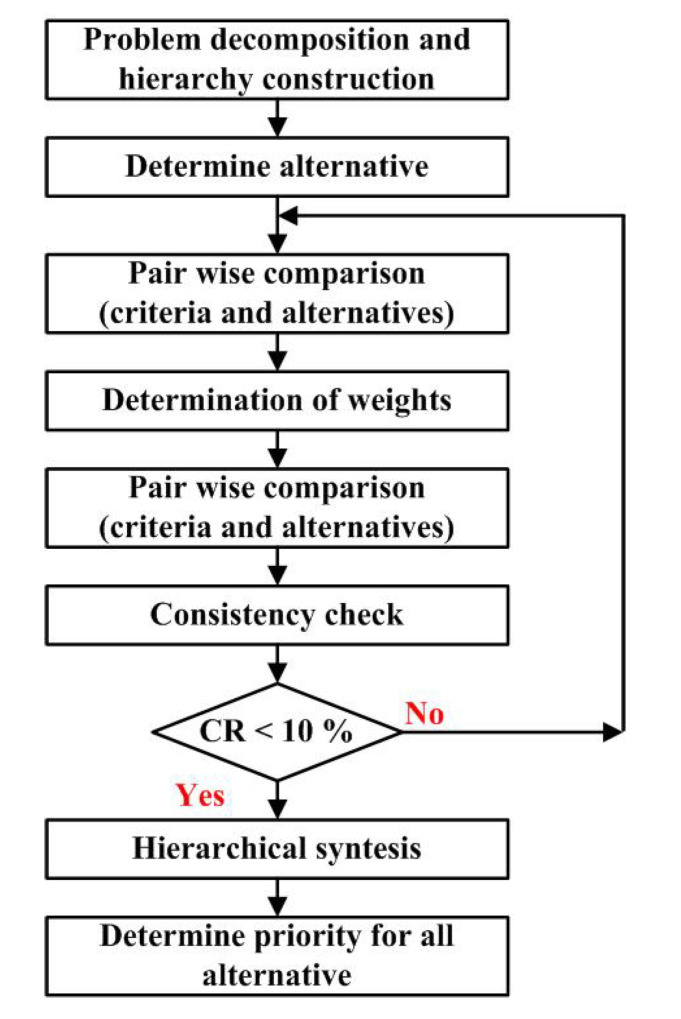
Analytical Hierarchy Process methodology [28].

**Figure 2 ijerph-18-01890-f002:**
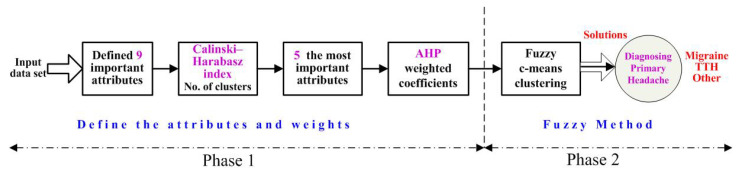
Flowchart of the Hybrid Intelligent System for Diagnosing Primary Headaches.

**Figure 3 ijerph-18-01890-f003:**
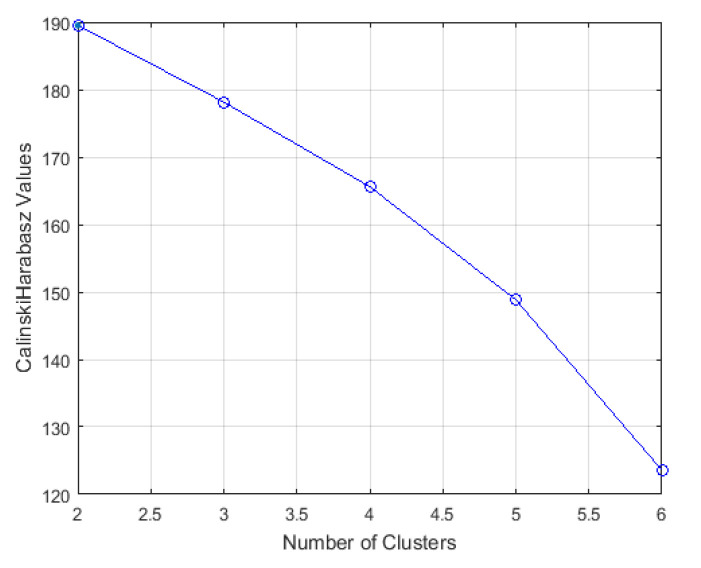
*Calinski-Harabasz Index* values for the estimation of the optimal number of clusters.

**Table 1 ijerph-18-01890-t001:** The International Classification of Headache Disorders–the Third Edition [21].

ICHD-3 Code			Diagnosis
**A**			**Primary Headache Disorders**
	**1.**		**Migraine**
		1.1	Migraine without aura (MWoA)
		1.2	Migraine with aura (MWA)
		¦	¦
		1.6	Episodic syndromes that may be associated with migraine
	**2.**		**Tension-type headache (TTH)**
		2.1	Infrequent episodic tension-type headache
		¦	¦
		2.4	Probable tension-type headache
	**3.**		**Trigeminal autonomic cephalalgias (TACs)**
	**4.**		**Other primary headache disorders**
**B**	**5.**		**Secondary headache disorders**
	¦		
	**12.**		

**Table 2 ijerph-18-01890-t002:** Comparison of selected attributes for primary headache on the basis of IHS diagnostic criteria: **1.** consistency measure filter, **2.** relief greedy, **3.** relief top10, **4.** genetic algorithm wrapper, **5.** ACO-based classification algorithm, **6.** rule-based fuzzy logic system, **7.** physician’s expert choice, **8.** column RES—final decision for the important attribute selection [22].

	Attributes	1	2	3	4	5	6	7	RES
1	Sex								
2	How old were you when the headache occurred for the first time?								
3	How often do you have headache attacks?								
4	How long do the headache attacks last?								
5	Where is the headache located?								
6	How intense is the pain?								
7	What is the quality of the pain you experience?								
8	Do your headaches worsen after physical activities such as walking?								
9	Do you avoid routine physical activities because you fear they might trigger your headache?								
10	Are the headaches accompanied by? a) Nausea								
11	Are the headaches accompanied by? b) Vomiting								
12	Are the headaches accompanied by? c) Photophobia								
13	Are the headaches accompanied by? d) Phonophobia								
14	Do you have temporary visual, sensory or speech disturbance?								
15	Do you, during a headache attack, have tension and/or heightened tenderness of head or neck muscles?							TTH	
16	Do you have any body numbness or weakness?								
17	Do you have any indications of oncoming headache?								
18	Headache is usually triggered by: Menstrual periods								
19	In the half or my visual field, lasting 5 minutes to an hour, along with the headache attack or an hour before.								
20	Along with the headache attack or an hour before one I have sensory symptoms.								

**Table 3 ijerph-18-01890-t003:** Confusion matrix for the multi-class problem.

		Actual Class		
**Predicted class**		**Migraine**	**TTH**	**Other**
	**Migraine**	TP_M_	E_TM_	E_OM_
	**TTH**	E_MT_	TP_T_	E_OT_
	**Other**	E_MO_	E_TO_	TP_O_

**Table 4 ijerph-18-01890-t004:** Matrix of Means and standard deviation values per classes.

Classes	Attributes								
	**(4)**	**(5)**	**(6)**	**(7)**	**(8)**	**(10)**	**(12)**	**(13)**	**(15)**
Migraine	**2.0 ± 0.2**	2.0 ± 1.0	**2.8 ± 0.6**	**1.3 ± 0.5**	**1.4 ± 0.5**	1.2 ± 0.4	1.3 ± 0.4	1.2 ± 0.4	**1.3 ± 0.4**
TTHs	**2.3 ± 0.6**	2.7 ± 1.0	**2.0 ± 0.6**	**1.8 ± 0.5**	**1.7 ± 0.5**	1.7 ± 0.5	1.7 ± 0.5	1.6 ± 0.5	**1.1 ± 0.4**
Other	**1.5 ± 0.8**	2.7 ± 0.7	**2.0 ± 0.7**	**1.7 ± 0.6**	**1.7 ± 0.4**	1.7 ± 0.5	1.6 ± 0.5	1.6 ± 0.5	**1.4 ± 0.5**

Bold numbers: the most important attributes are: A4, A6, A7, A8, and A15.

**Table 5 ijerph-18-01890-t005:** Pairwise comparison matrix.

	Attribute (4)	Attribute (6)	Attribute (7)	Attribute (8)	Attribute (15)
**Attribute (4)**	1	2	1	1	1
**Attribute (6)**	½	1	½	1	1
**Attribute (7)**	1	2	1	1	1
**Attribute (8)**	1	1	1	1	1
**Attribute (15)**	1	1	1	1	1

**Table 6 ijerph-18-01890-t006:** Confusion matrix for actual classes and predicted classes.

		Actual Classes			
		Migraine	TTHs	Other	Σ
**Predicted classes**	**Migraine**	146	44	27	217
	**TTH**	21	163	34	218
	**Other**	2	17	125	144
Total		169	224	186	579

**Table 7 ijerph-18-01890-t007:** Quality measurement for the Hybrid Intelligent System for Diagnosing Primary Headache.

	Migraine	TTHs	Other	Average
***Precision %***	67.3	74.8	86.8	76.3
***Recall %***	86.4	72.7	67.2	75.4
***F*** **_1_** ***score %***	75.6	73.7	75.7	75.0

## Data Availability

Data is contained within the article.

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
