# Peer review of "An Application of a Hybrid Intelligent System for Diagnosing Primary Headaches"

_ijerph, 2021, doi:10.3390/ijerph18041890_

Round 1
Reviewer 1 Report
The ms investigates the application of statistical approaches for diagnosing primary headaches. Ths ms is clearly structured and well written.
My comments are as follows:
1. L. 36: use no digit after decimal throughout the complete abstract.
2. L. 119: “usually“
3. L. 136ff.: maybe just use one digit after the decimal.
4. L. 166ff.. Numbers in parentheses seem to refer to Table 2. Maybe consider an abbreviation like A4, A5, …
5. Equation 1: Something must be incorrect here. What are the indices t, i, and j?
6. L. 271: use italic font for all mathematical symbols.
7. L. 310: what is E_XY. In Formula (4), there is E_ij? Does this mean the same symbol?
Author Response
Response to Reviewer 1 Comments
According to your suggestions we have tried to improve our paper in the following way.
The ms investigates the application of statistical approaches for diagnosing primary
headaches. Ths ms is clearly structured and well written.
Response: Authors are flattered and thankful for such a kind comment.
My comments are as follows:
1. L. 36: use no digit after decimal throughout the complete abstract.
Response 1: It is corrected, and now it is in line 35. Whole abstract is checked.
2. L. 119: “usually“
Response 2: It is corrected, and now it is in line 121.
3. L. 136ff.: maybe just use one digit after the decimal.
Response 3: It is corrected, and now from line 141 until 143.
4. L. 166ff.. Numbers in parentheses seem to refer to Table 2. Maybe consider an
abbreviation like A4, A5, …
Response 4: The abbreviation like A4, A5, ... now in whole manuscript is used. In lines 169,
337, from line 352 until line 356.
5. Equation 1: Something must be incorrect here. What are the indices t, i, and j?
Response 5: The Equation 1 is corrected, and described. In the expression (1), nt is the
number of elements that belongs to the cluster c, μ is the global centroid, μt is the centroid of
the cluster t.
=
= − −
C
t
t
T
t t B c n
1
( ) (μ μ ) (μ μ )
6. L. 271: use italic font for all mathematical symbols.
Response 6: Italic font is used for all mathematical symbols in whole manuscript. Now it is
in line 275.
7. L. 310: what is E_XY. In Formula (4), there is E_ij? Does this mean the same
symbol?
Response 7: (Now it is line 321) E_XY is not the same symbol as E_ij. E_ij depends from
the value when precision, recall are calculated. ETM, EOM, EMT are exactly value for concrete
problem presented in confusion matrix. For example, ETM, EOM are error values when
Predicted class is Migraine. But, on the other side ETM is error value when Actual class is
TTHs. That depends of the point of view in multi-class problem: true negative or false
positive. The detail discussion in multi-class problem is started in 2018.
We would like to express our deepest gratitude to the reviewers for their kind words and
support. Your contribution is very important to us, as well as rewarding.
Best Regards,
Prof. Dragan Simic
University of Novi Sad

Reviewer 2 Report
In this paper, the authors developed an application of a hybrid intelligent system for diagnosing primary headaches. This application uses various techniques including Calinski-Harabasz index, Analytical Hierarchy Process, and Weighted Fuzzy C-means Clustering Algorithm. As a result, they have succeeded in diagnosing primary headaches with high accuracy. So I recommend publication of this paper in this journal. The present version of this paper, however, needs an minor revision before publication.
- They did not discuss the characteristics of the case with the wrong diagnosis. They had better analyze the weaknesses of the application as well as its strengths.
Author Response
Response to Reviewer 2 Comments
According to your suggestions we have tried to improve our paper in the following way.
In this paper, the authors developed an application of a hybrid intelligent system for diagnosing primary headaches. This application uses various techniques including Calinski-Harabasz index, Analytical Hierarchy Process, and Weighted Fuzzy Cmeans Clustering Algorithm. As a result, they have succeeded in diagnosing primary headaches with high accuracy. So I recommend publication of this paper in this journal. The present version of this paper, however, needs an minor revision before publication.
Response: Authors express their gratitude for such kind words.
1. They did not discuss the characteristics of the case with the wrong diagnosis.
They had better analyze the weaknesses of the application as well as its strengths.- Response 1: In the manuscript, at the end of Section 4. are added two new paragraphs (from line 527 to 552) which are better describe weaknesses and strengths of the proposed intelligent system for diagnosing primary headaches. ------------------------------------------------------------ Weaknesses in diagnosis occur in everyday clinical work, since atypical disease forms exist, and such cases are certainly present in our research as well. Likewise, the characteristics of the disease - which are presented as attributes in this study, do not mean that each observed case meets all the characteristics in 100%, as established in The International Classification of Headache Disorders – Third Edition (ICHD-3) [21]. The difference in clinical presentations in some cases leads to deviations from the recommended criteria in ICHD-3, which is a weakness of the recommended hybrid intelligent system for diagnosing primary headaches. On the other hand, the patient may suffer from several types of headaches that occur at different times. Therefore, the obtained answers, i.e. attributes, can be a combination of two types of headaches, leading to discrepancies in the proposed
diagnosis provided by the intelligent system and the diagnosis by a neurologist, representing their expert assessment. The occurrence of two types of headaches with the same patient can be interpreted as the mean value of the observed attributes, having similar values for different types of headaches, as displayed in Table 4 for attributes A5, A10, A12, and A13 for TTHs and Other headache types.
The advantage of this system is the identification of the attributes that are dominant for different types of headaches. In this hybrid intelligent system for diagnosing primary headaches, their significance, in relation to other attributes, is increased by weighting coefficients in weighted fuzzy c-means clustering algorithm. The goal of our research is to help physicians have a reliable tool that would help them make the better diagnosis of primary headaches.
2. English language and style are fine/minor spell check required
Response 2: Full manuscript language edited by a native English speaker. We would like to express our deepest gratitude to the reviewers for their kind words and support. Thank you for helping us improve our paper and describe our research better.
Best Regards,
Prof. Dragan Simic
University of Novi Sad
